# UiO-66-NH_2_ and Zeolite-Templated Carbon Composites for the Degradation and Adsorption of Nerve Agents

**DOI:** 10.3390/molecules26133837

**Published:** 2021-06-23

**Authors:** Jaeheon Lee, Dongwon Ka, Heesoo Jung, Kyeongmin Cho, Youngho Jin, Minkun Kim

**Affiliations:** Agency for Defense Development, P.O. Box 35, Yuseong-gu, Daejeon 34186, Korea; jhun0740@gmail.com (J.L.); rkehd47@gmail.com (D.K.); hsjung@add.re.kr (H.J.); kmcho@add.re.kr (K.C.)

**Keywords:** UiO-66-NH_2_/ZTC composite, metal-organic framework, zeolite-templated carbon, nerve agent, soman (GD), sarin (GB)

## Abstract

Composites of metal-organic frameworks and carbon materials have been suggested to be effective materials for the decomposition of chemical warfare agents. In this study, we synthesized UiO-66-NH_2_/zeolite-templated carbon (ZTC) composites for the adsorption and decomposition of the nerve agents sarin and soman. UiO-66-NH_2_/ZTC composites with good dispersion were prepared via a solvothermal method. Characterization studies showed that the composites had higher specific surface areas than pristine UiO-66-NH_2_, with broad pore size distributions centered at 1–2 nm. Owing to their porous nature, the UiO-66-NH_2_/ZTC composites could adsorb more water at 80% relative humidity. Among the UiO-66-NH_2_/ZTC composites, U_0.8_Z_0.2_ showed the best degradation performance. Characterization and gas adsorption studies revealed that beta-ZTC in U_0.8_Z_0.2_ provided additional adsorption and degradation sites for nerve agents. Among the investigated materials, including the pristine materials, U_0.8_Z_0.2_ also exhibited the best protection performance against the nerve agents. These results demonstrate that U_0.8_Z_0.2_ has the optimal composition for exploiting the degradation performance of pristine UiO-66-NH_2_ and the adsorption performance of pristine beta-ZTC.

## 1. Introduction

Chemical warfare agents (CWAs) such as chlorine, mustard, VX, sarin (*O*-isopropyl methylphosphonofluoridate, GB), and soman (*O*-pinacolyl methylphosphonofluoridate, GD) were first used in World War I and have continued to be used for chemical attacks [1,2,3]. For example, nerve agents were stockpiled during the Cold War and were used in the Iran-Iraq War in the 1980s and attacks in Syria in 2017 [2]. The nerve agents GB and GD, which are both organophosphates with similar chemical structures, can deactivate enzymes and disrupt the nervous system [3]. Activated carbon has been commonly used to provide protection against CWAs. Nowadays, many researchers have synthesized porous adsorbents such as metal-organic frameworks (MOFs) and Zr(OH)_4_ to decompose and adsorb CWAs [4]. MOFs such as NU-1000, UiO-66, UiO-67, MIL-101(Cr), and HKUST-1 have narrow micropores that can enhance intermolecular interactions and thus improve the adsorption performance [5]. Among MOFs, Zr-based MOFs are known to destroy CWAs due to their high porosity and chemical stability [6,7,8,9,10,11,12]. In particular, UiO-66-NH_2_ has been frequently used for the degradation of nerve agents [7,13,14,15]. However, the narrow micropores of MOFs may interrupt mass transfer and hinder adsorption [5,16,17]. Therefore, MOF-carbon composites have recently been studied to circumvent these problems [5,18,19].

Zeolite-templated carbons (ZTCs) are highly ordered microporous carbon materials [20,21,22,23] that have been used in many applications, such as hydrogen storage [24,25], methane storage [26,27], CO_2_ capture [28], and catalysis [29]. The structures of ZTCs depend on the synthesis conditions and can be classified into three types by quality. Type-I ZTCs consist carbon of zeolite-templated three-dimensionally ordered frameworks only, which can be fabricated using BEA (beta), EMT, and FAU zeolites as templates [22,30]. Type-II ZTCs contain a mixture of the Type-I and non-templated carbons, and Type-III ZTCs rarely contain Type-I structure [22]. Among Type-I ZTCs, beta-ZTCs have recently been shown to have applicability as gas sorption materials, gas separation materials, and capacitors [30,31].

Herein, we synthesized UiO-66-NH_2_/ZTC composites for the adsorption and decomposition of GB and GD. The addition of beta-ZTC to UiO-66-NH_2_, which contains highly porous and highly ordered features, provided additional adsorption and degradation sites for nerve agents (Scheme 1). To the best of our knowledge, the application of ZTCs and UiO-66-NH_2_/ZTC composites as reactive adsorbents for nerve agents has not been previously reported.

## 2. Results and Discussion

### 2.1. Characterization of UiO-66-NH_2_/ZTC Composites and Pristine Materials

Pristine UiO-66-NH_2_ and UiO-66-NH_2_/ZTC composites were prepared by a solvothermal method with different ratios of UiO-66-NH_2_ and beta-ZTC. Materials with UiO-66-NH_2_/ZTC ratios of 2:8 and 8:2 were labeled as U_0.2_Z_0.8_, and U_0.8_Z_0.2_, respectively. Scanning electron microscopy (SEM) images of the UiO-66-NH_2_/ZTC composites and pristine materials are presented in Figure 1a–d, and the corresponding energy-dispersive X-ray spectroscopy (EDS) elemental analysis results are presented in Appendix A. As shown in Figure 1a, UiO-66-NH_2_ consists of particles with a size range of 100–300 nm. It has been reported that the particle size of UiO-66-NH_2_ depends on the ratio of hydrochloric acid (HCl) to *N*,*N*-dimethylformamide (DMF) during synthesis, with small particles being obtained when a high concentration of HCl is used as a modulator [13]. Irregular round-shaped with the size of around 200 nm of beta-ZTC is shown in Figure 1b, and transmission electron microscopy (TEM) images of beta-ZTC presented in Appendix A indicate an ordered carbon structure [32]. As shown in Figure 1c, for U_0.2_Z_0.8_, the growth of small UiO-66-NH_2_ particles occurred on the surface of beta-ZTC. It has been reported that MOFs are located in the pores or void spaces of activated carbon in MOF-carbon composites [33]. However, in U_0.2_Z_0.8_, the UiO-66-NH_2_ nanoparticles were dispersed on the surface of beta-ZTC, likely because beta-ZTC has micropores with sizes between 0.7 and 1.0 nm. Furthermore, UiO-66-NH_2_ was densely agglomerated around the beta-ZTC particles. In contrast, as the ratio of UiO-66-NH_2_ increased, UiO-66-NH_2_ particles were also synthesized between the beta-ZTC particles, and UiO-66-NH_2_ and beta-ZTC were uniformly dispersed in U_0.8_Z_0.2_ (Figure 1d). The EDS elemental analysis showed the correlation between the ratio of UiO-66-NH_2_ in materials and the ratio of Zr, N, and O in materials (Appendix A).

The attenuated total reflectance Fourier transform infrared (ATR-FTIR) spectra of the materials are presented in Figure 2a, and the powder X-ray diffraction (PXRD) patterns of the materials are presented in Figure 2b. In the ATR-FTIR spectra (Figure 2a), the bands at 1400–1767 cm^−1^ were ascribed to the carboxylic functional groups of UiO-66-NH_2_. In particular, the peak at 1658 cm^−1^ was attributed to C=O stretching, those at 1497 and 1573 cm^−1^ to OCO asymmetrical stretching, and those at 1385 and 1430 cm^−1^ to OCO symmetrical stretching [13,34,35,36]. The peak observed at 1258 cm^−1^ was assigned to the C–N stretching of UiO-66-NH_2_. The bands at 3367 and 3480 cm^−1^ were ascribed to the symmetric and asymmetric vibrations of -NH_2_ groups, which showed synergistic effects with the broad stretching vibration bands of -OH groups at 3000–3580 cm^−1^ [35,36]. The intensities of these peaks decreased as the ratio of UiO-66-NH_2_ decreased in the UiO-66-NH_2_/ZTC composites. The PXRD patterns of UiO-66-NH_2_ and beta-ZTC (Figure 2b) were consistent with reported data [22,34]. Among the UiO-66-NH_2_/ZTC composites, the PXRD pattern of U_0.2_Z_0.8_ shows that UiO-66-NH_2_ did not synthesize properly. This result can be confirmed in another analysis. In Appendix A, the EDS elemental analysis data also show a lower Zr ratio than the theoretical amounts that U_0.2_Z_0.8_ should have. The amount of Zr should have been 12.48%, theoretically, but only 6.31% of Zr was observed in U_0.2_Z_0.8_. On the other hand, the ATR-FTIR and PXRD patterns of U_0.8_Z_0.2_ show that the chemical and crystallographic properties of both pristine materials were successfully preserved after synthesis step. Furthermore, the EDS analysis showed that U_0.8_Z_0.2_ contains in 51.15% of Zr, which is similar to the predicted results of 49.91%. From these analytical results, two different substances were successfully incorporated into the composite U_0.8_Z_0.2_.

The thermogravimetric analysis (TGA) curves and dynamic vapor sorption (DVS) curves of the materials are presented in Figure 3a,b respectively. As shown in the TGA curves (Figure 3a), the ZTC, which is mainly carbonaceous, exhibited a weight loss above 500 °C. The thermal decomposition of that carbon materials can be expected from the intrinsic character of the ZTC. Guan et al., reported that Type-II ZTC can be thermally decomposed at 500 °C, even below 250 °C, due to their amorphous nature [37]. In contrast, the materials that included UiO-66-NH_2_ showed weight losses at 30–100 °C owing to the vaporization of water, at 250–300 °C owing to the dihydroxylation of the cluster, and at ~450 °C owing to the decomposition of the organic moieties [34,38,39]. As UiO-66-NH_2_ is strongly hydrophilic, the weight loss below 100 °C increased as the UiO-66-NH_2_ ratio increased.

These effects were also apparent in the DVS curves (Figure 3b). Pristine beta-ZTC exhibited a typical type V isotherm, whereas UiO-66-NH_2_ exhibited a typical type IV isotherm, as defined by the International Union of Pure and Applied Chemistry (IUPAC) classification [40]. At 50% relative humidity (RH), UiO-66-NH_2_ showed high water vapor adsorption, but beta-ZTC contained almost no water vapor. However, at 60% RH, beta-ZTC adsorbed water vapor rapidly. Although beta-ZTC is highly hydrophobic, it can adsorb more water vapor when the RH is high because of its numerous pores [41].

The nitrogen adsorption isotherms at 77 K and the pore size distributions of the UiO-66-NH_2_/ZTC composites and pristine materials are presented in Figure 4a–c. The Brunauer-Emmett-Teller (BET) surface areas (S_BET_) and total pore volumes at a relative pressure of 0.95 (V_T_) are summarized in Table 1.

The nitrogen adsorption isotherms of all the materials had the characteristics of typical type II isotherms, indicating the presence of a large number of micropores [42,43]. The BET specific surface area of pristine UiO-66-NH_2_ was 1105 m^2^/g with a total pore volume of 0.56 cm^3^/g. The pore size distribution calculated using density functional theory (DFT), indicated that pristine UiO-66-NH_2_ had pores with sizes of 0.6, 1.2, and 1.4 nm (Figure 4b). Based on the characterization results, medium to highly defective UiO-66-NH_2_ was synthesized, as reported in the literature [13].

The BET specific surface area of pristine beta-ZTC was 2560 m^2^/g with a total pore volume of 1.39 cm^3^/g. Furthermore, the pore size distribution demonstrated that pristine beta-ZTC mostly contained pores with a size of 1.12 nm as well as several pores with sizes around 0.8, 1.2, and 1.5 nm (Figure 4b). These characteristics coupled with the PXRD pattern (Figure 2b) and TEM images of beta-ZTC (Appendix A) demonstrated that beta-ZTC was between Type-I and Type-II, as reported in the literature [22].

The BET specific surface areas of the UiO-66-NH_2_/ZTC composites U_0.2_Z_0.8_ and U_0.8_Z_0.2_ were 2274 and 1360 m^2^/g, respectively, with total pore volumes of 1.22 and 0.65 cm^3^/g, respectively. With the increase in the beta-ZTC content, the specific surface area and total pore volume increased. The broad pore size distributions showed that the UiO-66-NH_2_/ZTC composites had pores with sizes centered around 1.12 nm. The latter pore, which are not found in the pristine materials, may improve the adsorption ability of UiO-66-NH_2_ (Figure 4c) [7,44,45,46].

### 2.2. Degradation Rates and Reaction Products of Nerve Agents

To analyze the performance of the UiO-66-NH_2_/ZTC composites and pristine materials for nerve agent degradation and the resulting reaction products, the materials were reacted with GB and GD in pentane. The reacted samples were then extracted with ethyl acetate and analyzed by gas chromatography–mass spectrometry (GC-MS).

During degradation, GB is first hydrolyzed to isopropyl methylphosphonoic acid (IMPA) and GD to pinacolyl methylphosphonoic acid (PMPA). IMPA and PMPA are then further hydrolyzed to methylphosphonoic acid (MPA), as described in Figure 5a [47]. Many researchers have shown that UiO-66-NH_2_ can degrade GD [7,13,14,15,45,48]; however, only the degradation of DMMP, a simulant for GB, has been previously reported [8,46]. Figure 5b,c show the degradation rates of GB and GD with pristine beta-ZTC, pristine UiO-66-NH_2_, U_0.2_Z_0.8_, and U_0.8_Z_0.2_. UiO-66-NH_2_ decomposed 90% of GB on average in 10 min (Figure 5b), and this is the first report of the GB degradation performance of UiO-66-NH_2_. UiO-66-NH_2_ also decomposed GD (Figure 5c), as previously reported. In contrast, little decomposition of GB and GD was observed with beta-ZTC. To detect the reaction products using GC-MS, the solutions were treated with *N*,*O*-bis(trimethylsilyl)trifluoroacetamide, which resulted in the hydrogen atoms of the hydroxyl groups being replaced with trimethylsilyl (TMS) groups [49,50]. Appendix A show the GC spectra and Appendix A–c show the MS spectra of the reaction products from GB and GD degradation. The degradation rates of the nerve agents were found to increase as the ratio of UiO-66-NH_2_ in the UiO-66-NH_2_/ZTC composites increased. According to the reaction product analysis (Figure 5), U_0.8_Z_0.2_ decomposed GB into IMPA and GD into PMPA, both of which eventually produced MPA, confirming the ability of the UiO-66-NH_2_/ZTC composites to degrade nerve agents.

### 2.3. Adsorption of Nerve Agents

In the previous section, the performance of the UiO-66-NH_2_/ZTC composites for nerve agent degradation was revealed. UiO-66-NH_2_ can provide sites for both adsorption and decomposition via hydrolysis [13,49]. To clarify the effect of beta-ZTC in the U_0.8_Z_0.2_ composites, a gas adsorption test was performed.

Figure 6a,b show the results of the GB and GD gas adsorption tests with pristine materials and U_0.8_Z_0.2_. Pristine beta-ZTC showed the best adsorption performance, with no nerve agents detected at the outlet. The standard peak of each nerve agent was also detected by micro-pulsed testing without the materials. The highest peaks for standard GB and GD were 401.2 and 71.7, respectively. Based on the highest peaks, U_0.8_Z_0.2_ can adsorb 1.5 times more GB and GD than pristine UiO-66-NH_2_ despite containing only 20% beta-ZTC. The adsorption performance and pore size distribution results (Figure 4) reveal that pristine beta-ZTC and U_0.8_Z_0.2_ have excellent adsorption abilities for GB and GD [49].

### 2.4. Protection Performance against GD

To investigate the synergic effect of the degradation and adsorption performance of the materials, a protection performance analysis was performed using the convective-flow swatch testing method and materials dip-coated in polyurethane foam.

Figure 7 shows the protection performance of the materials against GD. Convective-flow testing was performed to test the resistance of the materials to convective penetration by nerve agents [51]. U_0.8_Z_0.2_ exhibited greater protection against GD under convective conditions than the pristine materials and U_0.2_Z_0.8_, which indicates that the nerve agent was adsorbed and decomposed by UiO-66-NH_2_ and beta-ZTC. The characterization, degradation, adsorption, and protection performance results indicate that small amounts of beta-ZTC in the composite with UiO-66-NH_2_ can promote the formation of suitable pores and provide additional water-containing sites that can adsorb and decompose GB and GD more effectively owing to the even dispersion of the components. To the best of our knowledge, this work is the first to describe GB and GD adsorption and degradation by beta-ZTC and its composites with UiO-66-NH_2_.

## 3. Materials and Methods

### 3.1. Materials

Beta-ZTC was supplied by Prof. Kyoungsoo Kim from Jeonbook University. Zirconyl chloride octahydrate (ZrOCl_2_∙8H_2_O, 98%, powder), DMF (≥99.8%), HCl (37%), ethyl acetate (anhydrous, 99.8%), and *N*,*O*-bis(trimethylsilyl)trifluoroacetamide (≥99.0%) were purchased from Sigma-Aldrich (St. Louis, MO, USA). 2-Aminoterephtalic acid (BDC-NH_2_, 99%, powder) was purchased from Alfa Aesar (Haverhill, MA, USA). All chemicals were used as received without further treatment. GB and GD were synthesized at OPCW-designated laboratories. The purities of GD and GB, as determined by GC-MS and ^1^H NMR, exceeded 99%.

### 3.2. Preparation of UiO-66-NH_2_/ZTC Composites (U_x_Z_1−x_) and UiO-66-NH_2_

UiO-66-NH_2_/ZTC composites (U_x_Z_1−x_|UiO-66-NH_2_:ZTC = x:1 − x) and pristine UiO-66-NH_2_ were synthesized by a solvothermal method. HCl (35 mL) was added to DMF (465 mL). Then, 2.80× *g* of ZrOCl_2_∙8H_2_O and 1.575× *g* of BDC-NH_2_ were dissolved in the solution by sonication for 1 h. Subsequently, 2.0(1 − x) g of beta-ZTC was dissolved in the solution by sonication for 1 h. The solution was allowed to react at 120 °C for 24 h. After cooling to room temperature, the resulting product was filtered, washed three times with DMF and three times with ethanol, and dried at 120 °C for 12 h. UiO-66-NH_2_ was synthesized similarly without the addition of beta-ZTC.

### 3.3. Characterization

SEM and EDS analyses were performed on a JSM-IT500HR scanning electron microscope (JEOL, Tokyo, Japan) equipped with an Xplore EDS detector (Oxford Instruments, Abingdon, UK) with acceleration voltages ranging from 5 to 10 kV. The materials were coated with platinum prior to SEM observation. PXRD patterns were obtained using a SmartLab diffractometer (Rigaku, Tokyo, Japan). ATR-FTIR spectra were acquired using a Nicolet iS50 spectrometer (Thermo Fisher Scientific, Waltham, MA, USA). TGA measurements were performed using a Q500 analyzer (TA Instruments, New Castle, PA, USA) by heating the sample from 30 to 800 °C at a rate of 10 °C/min under nitrogen gas. DVS measurements were performed at 25 °C using a DVS Intrinsic analyzer (Surface Measurement Systems, London, UK). Nitrogen adsorption measurements were performed at 77 K using an ASAP 2020 system (Micromeritics Instrument Corp., Norcross, GA, USA). Before analysis, the samples were degassed at 393 K for 2 h under vacuum. The nitrogen isotherm data were used to calculate the specific surface area using the BET method and the pore size distribution using a DFT model.

### 3.4. Degradation of Nerve Agents and Reaction Products Analysis

The as-synthesized materials were exposed to 80% RH at 25 °C for 24 h and then subdivided into 2 mL vials (10 mg each). Subsequently, 0.4 μL of nerve agent in 20 μL of pentane was added to each vial. After agitation for 10 min on a vortex mixer, the residual nerve agent was extracted with ethyl acetate (1.5 mL) for 2 h. The solution was derivatized with *N*,*O*-bis(trimethylsilyl)trifluoroacetamide at 70 °C for 2 h [52]. Both extracted and derivatized solutions were analyzed by GC-MS (GC 7890A, MSD 5977A, Agilent Technologies, Madison, WI, USA) [47].

### 3.5. Analysis of Adsorption Performance against Nerve Agents

A micro-pulsed testing method was used to analyze the adsorption performance of the materials (Scheme 2). This method is based on the micro-breakthrough testing method and the liquid challenge/vapor penetration (L/V) swatch testing method [51,53]. A glass tube with a nominal inner diameter of 4 mm was packed with 20 mg of material, which was fixed in place with glass fiber at the top and bottom. Then, 8 μL of nerve agent was injected and the concentration of the nerve agent at the outlet was observed using GC-FID (GC 7890A, Agilent Technologies, Madison, WI, USA) at a flow rate of 50 mL/min.

### 3.6. GD Protection Performance Analysis (Swatch Testing)

To analyze the GD protection performance, the convective-flow-liquid challenge/vapor penetration (L/V) swatch testing method from TOP-08-2-501A was used [51]. Convective-flow testing was performed to test the resistance of the materials to convective penetration by nerve agents. The as-synthesized materials were dip-coated in polyurethane foam (Testfabrics, West Pittston, PA, USA) and cut into pieces with a diameter of 45 mm. The cut pieces were treated in a temperature humidity chamber at 32 °C and 80% RH for ≥2 h. After mounting in the test cell, the material was covered with 10 drops (1 μL each) of GD and the test cell was assembled. For convective-flow testing, the airflow rate was controlled to maintain a ΔP value of 24.9 ± 3.7 Pa between the top and bottom of the test cell (Scheme 3). Air from the underside of the test cell was collected using a Porapak Q adsorption tube. Compounds were desorbed from the tubes using a thermal desorption instrument (Ultra-Unity, Markes, Llantrisant, UK) and analyzed using GC-FID (GC 7890A, Agilent Technologies, Madison, WI, USA).

## 4. Conclusions

This study evaluated the performance of UiO-66-NH_2_/ZTC composites and the corresponding pristine materials for the adsorption and decomposition of nerve agents. The composites had higher specific surface areas than pristine UiO-66-NH_2_. Furthermore, the composites had broad pore size distributions with new larger pores, which can absorb more water at 80% RH. The degradation rates of GB and GD increased as the ratio of UiO-66-NH_2_ in the composite increased. In contrast, the gas adsorption ability increased as the ratio of beta-ZTC increased. Furthermore, convective-flow swatch testing revealed a synergistic effect between the degradation and adsorption performance. Among the investigated materials, U_0.8_Z_0.2_ showed the best protection performance against nerve agents. The improved performance of the U_0.8_Z_0.2_ composite was attributed to the presence of the optimal amount of beta-ZTC, which can provide additional adsorption and degradation sites for nerve agents.

## Data Availability

The data presented in this study are available within the article or Appendix A.

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
