# Peer review of "UiO-66-NH2 and Zeolite-Templated Carbon Composites for the Degradation and Adsorption of Nerve Agents"

_molecules, 2021, doi:10.3390/molecules26133837_

Round 1
Reviewer 1 Report
In the study, the author synthesized series of UiO-66-NH2/zeolite-templated carbon composites for the adsorption and decomposition of the nerve agents sarin and soman. The results showed that the composites exhibited greater degradation performance than the corresponding counterparts. In my opinion, this work is interesting and could be accepted after correcting the following problems.
- The word “since“ in line 28 should be deleted.
- The second “and” in line 61 should be deleted.
- In figure 1, UiO-66-NH2 and its composites with ZTC carbon were analyzed. But carbon in Figure 1b was missed. The author should make a supplement.
- The author mentioned Table S1 in line 70, but no analysis was performed. Beside, Figure S1 was given in the supporting information, but no correlated content was given in the manuscript.
- In line 141, demonstrate should be corrected into demonstrated.
- In line 164, the author stated that over 85% of GB was decomposed by UiO-66-NH2 in 10 min. However, the data read from Fig.5b seems to be 90%. The author should check it again.
- In Figure 3a, ZTC, which is a carbon material, showed a big weight loss above 500 oC. What is the reason for this loss?
- In 3.4 part, the materials were firstly exposed to 80% RH to adsorb some water, and then were used to test the degradation of nerve agents. From Figure 3b, it could be clearly seen that ZTC exhibited the best adsorption performance for water at 80% RH. And from Figure 5a, we could see that water plays an important role in the hydrolysis of GB and GD. So, considering these, ZTC should exhibit the best degradation performance of GB and GD. However, the fact is that ZTC is the worst in these investigated materials shown in Figure 5b and c. Why?
- In terms of MOF composites, I suggest the author could cite the following articles: Zhang, et al. Adsorption 2015, 21: 77-86; Z. Zhang, et al. RSC Advances 2018, 8, 21460.
Author Response
Reviewer 1:
General comment:
In the study, the author synthesized series of UiO-66-NH2/zeolite-templated carbon composites for the adsorption and decomposition of the nerve agents sarin and soman. The results showed that the composites exhibited greater degradation performance than the corresponding counterparts. In my opinion, this work is interesting and could be accepted after correcting the following problems.
Our response:
We truly appreciate the referee for his or her consideration on our article and insightful comments. In accordance with the comments, we have revised the manuscript and supplementary information in point-by-point manner.
Comment #1
The word “since” in line 28 should be deleted.
Our response #1
Thank you for your kindness comments. The comments you suggested #1 and #2 have made the sentences more valuable. We have changed that text based on your comments and reflected it in the manuscript.
Comment #2
The second “and” in line 61 should be deleted.
Our response #2
Thank you for your kindness comments. The comments you suggested #1 and #2 have made the sentences more valuable. We have changed that text based on your comments and reflected it in the manuscript.
Comment #3
In figure 1, UiO-66-NH2 and its composites with ZTC carbon were analyzed. But carbon in Figure 1b was missed. The author should make a supplement.
Our response #3
Thank you for your valuable comments. It seems like that more detailed explanation for ZTC is needed. Based on your comments, we have added the following sentences and supplementary reference.
- Our modification to the manuscript:
It has been reported that the particle size of UiO-66-NH2 depends on the ratio of hydrochloric acid (HCl) to N,N-dimethylformamide (DMF) during synthesis, with small particles being obtained when a high concentration of HCl is used as a modulator [13]. Irregular round-shaped with the size of around 200 nm of beta-ZTC is shown in Figure 1b, and transmission electron microscopy (TEM) images of beta-ZTC presented in Figure S1 indicate an ordered carbon structure [32].
Comment #4
The author mentioned Table S1 in line 70, but no analysis was performed. Beside, Figure S1 was given in the supporting information, but no correlated content was given in the manuscript.
Our response #4
Thank you for your insightful comments and we attached additional explanation for EDS analysis in manuscript. We sincerely apologized that some corrections were made to the results of EDS data, inevitably.
At first, the yield of the MOF was calculated to 78.7%. The theoretical total amount of UiO-66-NH2 can be induced from our experiment. As we used 8.7 mmol of ZrOCl2ž8H2O and BDC-NH2, theoretical amount of UiO-66-NH2 [Zr6O4(OH)4(C6H3(NH2)(COO)2)6]n is 2.543 g. But considering that weight of the final product is 2.0g, we can conclude that the yield of MOF is 78.7%. To synthesize the composites with a ratio of 8:2 and 2:8, the concentration of precusor is controlled based on the yield of MOFs. Proper composites designed in advance were obtained from the combination of calculated yield and controlling concentration of precusor. The ratio of prepared composite materials was determined via EDS analysis. The ratio of zirconium in pure UiO-66-NH2 is 62.39%, while carbon ratio origined from organic linker is 22.07%. Therefore, U0.8Z0.2, in which 80% of the composites is composed of MOFs, can theoretically contain 49.91% of the zirconium. For carbon element, 20% of carbon in ZTC and 80% of carbon in UiO-66-NH2 affected on the total carbon amounts of the U0.8Z0.2. In the same way as calculating zirconium, the total carbon amounts in U0.8Z0.2 is calculated to be 36.54%. we can obtained similar results of real amounts of each elements from EDS analysis. it was confirmed that U0.8Z0.2 contained 51.15% of zirconium and 33.99% of carbon in real state.
However, when calculating in the same way, U0.2Z0.8 leads to slightly different results. The amount of zirconium, 6.31%, observed from the EDS analysis is less than that of exected. From this result, we can consider that the composite may not have been synthesized properly, which was confirmed through the result that the peak of UiO-66-NH2 was hardly observed in the XRD data in figure 2b. Since the UiO-66-NH2 was not crystallized properly in the synthesis step, the zirconium elements can be washed and only a very small amounts of zirconium was remained.
And we have revised the sentence for figure S1 according to your suggestion.
- Our modification to the manuscript 1:
At first, we added the text below in line 85.
In contrast, as the ratio of UiO-66-NH2 increased, UiO-66-NH2 particles were also synthesized between the beta-ZTC particles, and UiO-66-NH2 and beta-ZTC were uniformly dispersed in U0.8Z0.2 (Figure 1d). The EDS elemental analysis showed the correlation between the ratio of UiO-66-NH2 in materials and the ratio of Zr, N, O in materials (Table S1).
To explain more detail, another sentences has been added in line 104-110.
Among the UiO-66-NH2/ZTC composites, PXRD pattern of U0.2Z0.8 shows that UiO-66-NH2 did not synthesized properly. This results can be confirmed in another analysis. In Table S1, the EDS elemental analysis data also shows lower Zr ratio than theoretical amounts that U0.2Z0.8 should have. The amount of Zr should have been 12.48% theoretically, but only 6.31% of Zr was observed in U0.2Z0.8. On the other hands, the ATR-FTIR and PXRD patterns of U0.8Z0.2 show that the chemical and crystallographic properties of both pristine materials were successfully preserved after synthesis step. Furthermore, the EDS analysis showed that U0.8Z0.2 contains in 51.15% of Zr which is similar to the predicted results of 49.91%. From these analytical results, two different substances were successfully incorporated into the composite U0.8Z0.2.
- Our modification to the manuscript 2:
It has been reported that the particle size of UiO-66-NH2 depends on the ratio of hydrochloric acid (HCl) to N,N-dimethylformamide (DMF) during synthesis, with small particles being obtained when a high concentration of HCl is used as a modulator [13]. Irregular round-shaped with the size of around 200 nm of beta-ZTC is shown in Figure 1b, and transmission electron microscopy (TEM) images of beta-ZTC presented in Figure S1 indicate an ordered carbon structure [32].
Comment #5
In line 141, demonstrate should be corrected into demonstrated.
Our response #5
Thank you again for your kindness comments following #1 and #2. We revised the sentence according to your opinion.
- Our modification to the manuscript:
These characteristics coupled with the PXRD pattern (Figure 2b) and TEM images of beta-ZTC (Figure S1) demonstrated that beta-ZTC was between Type-I and Type-II, as reported in the literature [22].
Comment #6
In line 164, the author stated that over 85% of GB was decomposed by UiO-66-NH2 in 10 min. However, the data read from Fig.5b seems to be 90%. The author should check it again.
Our response #6
Thank you for your valuable comments and we sincerely apologized to our uncleared expression. As you mentioned that decomposition rate was above 90% in average and error rage was ±1%. Actually, the demonstrated data in figure 5b, showed a higher level of degratdation rate than what we expressed in the text. We expressed a slightly lower level of degradation rate considering the error range, but we thought that it is more sensible to express the values given on the graph as suggested by refree. We have corrected “Over 85%” to “90% of GB on average”.
- Our modification to the manuscript:
Figure 5b and c show the degradation rates of GB and GD with pristine beta-ZTC, pristine UiO-66-NH2, U0.2Z0.8, and U0.8Z0.2. UiO-66-NH2 decomposed 90% of GB on average in 10 min (Figure 5b),
Comment #7
In Figure 3a, ZTC, which is a carbon material, showed a big weight loss above 500 oC. What is the reason for this loss?
Our response #7
Thank you for your insightful comments. As you know that, in inert gas condition, generally, crystalline graphite and fully graphitized carbon are not thermally decomposed at around 500℃. But in the case of amorphous carbon or carbons with functional groups or activated carbons, weight loss have been occurred during TGA analysis even low temperature. The ZTC reported in this paper was in Type-II ZTC which has highly amorphous structure. C. Guan et al. reported that Type-II ZTC can be thermally decomposed 500℃, even below 250℃, due to their amorphous nature.[C. Guan, et al. Microporous Mesoporous Mater. 2009, 118, 503-507)] According to the study, we can conclude that the thermal decomposition of the ZTC resulted from intrinsic property of that materials. We revised the manuscript as follows based on your comments.
- Our modification to the manuscript:
As shown in the TGA curves (Figure 3a), the ZTC, which is mainly carbonaceous, exhibited a weight loss above 500 °C. The thermal decomposition of that carbon materials can be expected from the intrinsic character of the ZTC. Guan et al. reported that Type-II ZTC can be thermally decomposed 500℃, even below 250℃, due to their amorphous nature[37].
[37] Guan, C.; Wang, K.; Yang, C.; Zhao, X. S. Characterization of a zeolite-templated carbon for H2 storage application. Microporous Mesoporous Mater. 2009, 118, 503-507.
Comment #8
In 3.4 part, the materials were firstly exposed to 80% RH to adsorb some water, and then were used to test the degradation of nerve agents. From Figure 3b, it could be clearly seen that ZTC exhibited the best adsorption performance for water at 80% RH. And from Figure 5a, we could see that water plays an important role in the hydrolysis of GB and GD. So, considering these, ZTC should exhibit the best degradation performance of GB and GD. However, the fact is that ZTC is the worst in these investigated materials shown in Figure 5b and c. Why?
Our response #8
Thank you for your valuable comments. We apologize for not being able to express a more detailed explanation. As you mentioned that nerve agents, such as GD, GB, are generally decomposed via hydrolysis mechanism. But under general condition, without any catalytic points, that reaction rate of hydrolysis is slightly low. A. F. Kingery et al. reviewed that the hydrolysis process of nerve agents based on organophosphates is affected by pH. For example, GB can be destructed in a few seconds in pH 12, but in the weak acid condtion or neutral condtion, their kinetic reaction progressed very slowly, even it take over a week to decompose in half. However, in the presence of catalysts, the hydrolysis reaction rate dramatically increases even in atmospheric condition. For these reason, UiO-66-NH2 can decompose the nerve agents relatively quickly, but the ZTC, with no catalytic site, hardly decomposed that nerve agents.
*[Kingery, A.F.; Allen, H.E. The envrionmental fate of organophosphorus nerve agents: a review. Toxicol Environ Chem. 1995, 47, 155-184]
Comment #9
In terms of MOF composites, I suggest the author could cite the following articles: Zhang, et al. Adsorption 2015, 21: 77-86; Z. Zhang, et al. RSC Advances 2018, 8, 21460.
Our response #9
Thank you for your mindful suggestion. From the papers you suggested, we found that they were not only conducted very insightful studies, but also proper contents that can make our manuscript more valuable were included. So, it makes our
paper more valuable when we site what you recommend.
- Our modification to the manuscript:
- Zhang, Z.; Wang, H.; Chen, X.; Zhu, C.; Wei, W.; Sun, Y. Chromium-based metal-organic framework/mesoporous carbon composite: synthesis, characterization and CO2 adsorption. Adsorption 2015, 21, 77-86.
- Zhang, Z.; Sun, N.; Wei, W.; Sun, Y. Facilely controlled synthesis of a core-shell structured MOF composite and its derived N-doped hierarchical porous carbon for CO2 adsorption. RSC Adv. 2018, 8, 21460.

Reviewer 2 Report
This paper presented the study on the fabrication of UiO-66-NH2/ZTC composites and their performance for the adsorption and decomposition of nerve agents, including GB and GD. There are both scientific and technological interests in the work. The comments were given as follows:
- The EDS elemental analysis result of ZTC should be provided and compared with the other three samples in Table S1. How was the ratio between the UiO-66-NH2 and ZTC in the composites measured? How much was the yield of the MOF?
- The error bars needs to be given in Fig 5b-c and Fig 7.
- Along with the TEM image of ZTC, it would be better to provide the TEM images of the composites to show the structure of MOF particles and their distribution on ZTC.
Author Response
Reviewer 2
General comment:
This paper presented the study on the fabrication of UiO-66-NH2/ZTC composites and their performance for the adsorption and decomposition of nerve agents, including GB and GD. There are both scientific and technological interests in the work. The comments were given as follows:
Our response:
We truly appreciate the referee for his or her consideration on our article and insightful comments. In accordance with the comments, we have revised the manuscript and supplementary information in point-by-point manner.
Comment #1
The EDS elemental analysis result of ZTC should be provided and compared with the other three samples in Table S1. How was the ratio between the UiO-66-NH2 and ZTC in the composites measured? How much was the yield of the MOF?
Our response #1
Thank you for your insightful comments and we attached additional explanation for EDS analysis in manuscript. We sincerely apologized that some corrections were made to the results of EDS data, inevitably. As you mentioned that EDS result of ZTC was added in the Table S1 and modified the incorrected results.
The yield of the MOF was calculated to 78.7%. The theoretical total amount of UiO-66-NH2 can be induced from our experiment. As we used 8.7 mmol of ZrOCl2ž8H2O and BDC-NH2, theoretical amount of UiO-66-NH2 [Zr6O4(OH)4(C6H3(NH2)(COO)2)6]n is 2.543 g. But considering that weight of the final product is 2.0g, we can conclude that the yield of MOF is 78.7%. To synthesize the composites with a ratio of 8:2 and 2:8, the concentration of precusor is controlled based on the yield of MOFs. Proper composites designed in advance were obtained from the combination of calculated yield and controlling concentration of precusor. The ratio of prepared composite materials was determined via EDS analysis. The ratio of zirconium in pure UiO-66-NH2 is 62.39%, while carbon ratio origined from organic linker is 22.07%. Therefore, U0.8Z0.2, in which 80% of the composites is composed of MOFs, can theoretically contain 49.91% of the zirconium. For carbon element, 20% of carbon in ZTC and 80% of carbon in UiO-66-NH2 affected on the total carbon amounts of the U0.8Z0.2. In the same way as calculating zirconium, the total carbon amounts in U0.8Z0.2 is calculated to be 36.54%. we can obtained similar results of real amounts of each elements from EDS analysis. it was confirmed that U0.8Z0.2 contained 51.15% of zirconium and 33.99% of carbon in real state.
However, when calculating in the same way, U0.2Z0.8 leads to slightly different results. The amount of zirconium, 6.31%, observed from the EDS analysis is less than that of exected. From this result, we can consider that the composite may not have been synthesized properly, which was confirmed through the result that the peak of UiO-66-NH2 was hardly observed in the XRD data in figure 2b. Since the UiO-66-NH2 was not crystallized properly in the synthesis step, the zirconium elements can be washed and only a very small amounts of zirconium was remained.
- Our modification to the manuscript:
At first, we added the text below in line 85.
In contrast, as the ratio of UiO-66-NH2 increased, UiO-66-NH2 particles were also synthesized between the beta-ZTC particles, and UiO-66-NH2 and beta-ZTC were uniformly dispersed in U0.8Z0.2 (Figure 1d). The EDS elemental analysis showed the correlation between the ratio of UiO-66-NH2 in materials and the ratio of Zr, N, O in materials (Table S1).
To explain more detail, another sentences has been added in line 104-110.
Among the UiO-66-NH2/ZTC composites, PXRD pattern of U0.2Z0.8 shows that UiO-66-NH2 did not synthesized properly. This results can be confirmed in another analysis. In Table S1, the EDS elemental analysis data also shows lower Zr ratio than theoretical amounts that U0.2Z0.8 should have. The amount of Zr should have been 12.48% theoretically, but only 6.31% of Zr was observed in U0.2Z0.8. On the other hands, the ATR-FTIR and PXRD patterns of U0.8Z0.2 show that the chemical and crystallographic properties of both pristine materials were successfully preserved after synthesis step. Furthermore, the EDS analysis showed that U0.8Z0.2 contains in 51.15% of Zr which is similar to the predicted results of 49.91%. From these analytical results, two different substances were successfully incorporated into the composite U0.8Z0.2.
Comment #2
The error bars needs to be given in Fig 5b-c and Fig 7.
Our response #2
Thank you for your kindful comments. We revised experimental data for the degradation of nerve agents and protective performance according to your suggestions. The modified part of our revised manuscript is as follows:
Figure 5. (a) Hydrolysis pathways of nerve agents. GC chromatograms and degradation rates for nerve agents (b) sarin (GB) and (c) soman (GD) with UiO-66-NH2/ZTC composites, pristine beta-ZTC, and pristine UiO-66-NH2.
Figure 7. Protection performance of UiO-66-NH2/ZTC composites, pristine beta-ZTC, and pristine UiO-66-NH2 against GD using the convective-flow swatch testing method.
Comment #3
Along with the TEM image of ZTC, it would be better to provide the TEM images of the composites to show the structure of MOF particles and their distribution on ZTC.
Our response #3
Thank you very much for your valuable comment. We apologize for not providing more detailed components. Following your insightful comment, we did try our best to carry out TEM measurement for structural analysis of our composite materials. Unfortunately, our institution has no other TEM machine used by ourselves and in addition, such processes for transferring our materials out of the institution take so many times over 1 week due to confidential issues. Besides, COVID-19 made us to fail to book reservation for TEM machines outside at the right deadline for our revision. We ask for your understanding that it was difficult to completely fulfill requested answer due to the unusual situation.